# Automatic correction of performance drift under acquisition shift in medical image classification

Mélanie Roschewitz [1,2] ✉, Galvin Khara[1], Joe Yearsley[1], Nisha Sharma[3], Jonathan J. James[4], Éva Ambrózay[5], Adam Heroux[1], Peter Kecskemethy[1], Tobias Rijken [1] & Ben Glocker [1,2] ✉

Image-based prediction models for disease detection are sensitive to changes in data acquisition such as the replacement of scanner hardware or updates to the image processing software. The resulting differences in image characteristics may lead to drifts in clinically relevant performance metrics which could cause harm in clinical decision making, even for models that generalise in terms of area under the receiver-operating characteristic curve. We propose Unsupervised Prediction Alignment, a generic automatic recalibration method that requires no ground truth annotations and only limited amounts of unlabelled example images from the shifted data distribution. We illustrate the effectiveness of the proposed method to detect and correct performance drift in mammography-based breast cancer screening and on publicly available histopathology data. We show that the proposed method can preserve the expected performance in terms of sensitivity/specificity under various realistic scenarios of image acquisition shift, thus offering an important safeguard for clinical deployment.

Artificial intelligence (AI) holds the promise for more objective, accurate, and cost-effective analysis of imaging data and could fundamentally transform clinical workflows in image-based diagnostics and population screening. The use of AI could help to ease the pressure on health services, for example, through automated prioritisation of critical cases[1,2], or by providing second opinions in diagnostic screening[3,4].

Different use cases will have different requirements on AI performance, depending on the role of the AI system within the clinical workflow. An AI system used for triaging or prioritisation would be expected to have near-perfect sensitivity (SEN), while a low specificity (SPC) may be acceptable. AI as a second reader in double reading breast cancer screening, on the other hand, may be expected to perform at similar SEN/SPC levels as a human reader. Many AI systems are versatile prediction models that can be used at different operating points in terms of a specific clinically meaningful SEN/SPC trade-off. Here, an operating point is associated with a calibrated threshold on the continuous AI prediction score. The thresholds are often predetermined, ideally as part of large-scale validation studies. Thresholds may also be adjusted as part of local optimisation using historical data from a new deployment site[5]. Here, the assumption is that such validation data is, and remains, representative of new data from unseen patients and the associated operating point is within approved values.

Changes occurring after deployment pose a fundamental challenge to medical imaging AI[6–10]. Current systems are susceptible to differences in the image characteristics. Changes to the input images, for example, due to replacement of scanner hardware or updates to the image reconstruction and processing software, can cause differences in the AI output predictions (see Fig. 1). These changes may occur any time after deployment and are often outside of the control

[1]Kheiron Medical Technologies, London, UK. [2]Imperial College London, Department of Computing, London, UK. [3]Leeds Teaching Hospital NHS Trust, Department of Radiology, Leeds, UK. [4]Nottingham University Hospitals NHS Trust, Nottingham City Hospital, Nottingham Breast Institute, Nottingham, UK. [5]MaMMa Egészségügyi Zrt., Budapest, Hungary. ✉e-mail: mb121@imperial.ac.uk; b.glocker@imperial.ac.uk

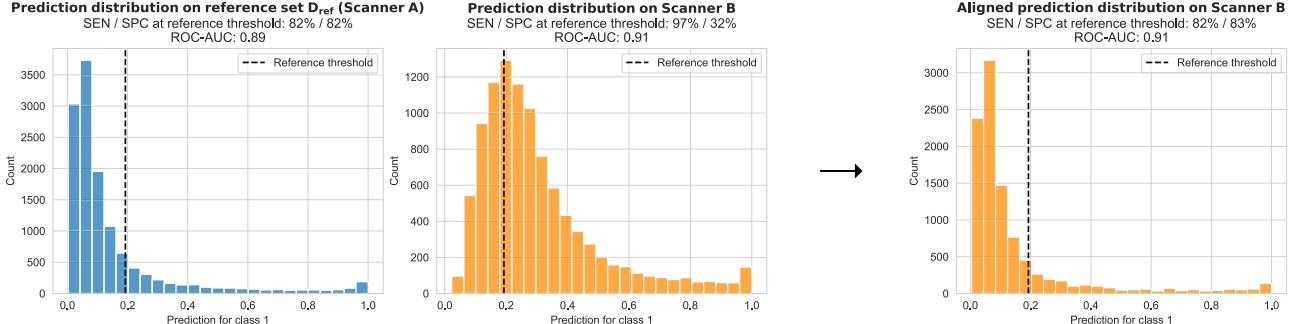

**Fig. 1 | Illustration of a shift in model prediction when applying a mammography malignancy detection model to a new scanner.** From left to right: **a** distribution of predictions observed on the reference set coming from a scanner representative of the majority of the development set and used to set the classification threshold, **b** observed distribution of predictions when applying this model on a completely unseen scanner we can observe that despite ROC-AUC generalisation there is a clear shift in model predictions and calibration, **c** effect of applying the proposed UPA method in terms of predictions distribution and SEN/SPC trade-off. Source data are provided as a Source Data file.

of the AI manufacturer. While changes in the image characteristics may not necessarily affect the overall predictive power (i.e., the ability to discriminate between positive and negative cases), any shift or drift in the output distribution may invalidate the calibrated thresholds. This would directly impact the SEN/SPC trade-off, and thus, the intended operating point of the AI system. If undetected, such clinical performance drift could result in under- or overdiagnosis and have severe consequences on patient safety.

There is an urgent need for an effective methodology for monitoring AI in real-world deployments, enabling automated correction of performance drift that is caused by image acquisition shifts. The topic of performance monitoring is a pressing matter for the practical deployment of machine learning models, in particular in critical applications such as disease detection[11,12]. In the past, some methods have been proposed to detect such performance drifts and update models[13–18], but these all require human annotations of new samples. However, obtaining annotations in near real-time and linking them with the acquired images is usually not possible in clinical practice, in particular in applications such as breast cancer screening where confirmed diagnosis (e.g., biopsy-proven malignancy) is not available at the time of image analysis. Some methods have been proposed to automatically detect shifts based on statistical tests comparing model predictions[19], but there remains an unmet need for methods allowing to automatically *correct* performance drift in the absence of any diagnostic information (such as disease labels). This is precisely the focus of this work.

In this work, we propose and evaluate a simple, generic, and effective approach of unsupervised prediction alignment (UPA) which is capable of detecting and correcting AI clinically relevant performance drift caused by acquisition shift. Our experiments demonstrate the effectiveness of UPA in several real-world scenarios in the context of breast screening and histopathology. In the breast screening application, we first show that UPA is able to adapt outputs of a model optimised on one hardware vendor to recover the desired SEN/SPC performance on different vendors across three large-scale UK breast screening datasets. We confirm the generic nature of UPA using the publicly available WILDS Camelyon17[20] dataset showing that a classification model optimised on a particular staining protocol can automatically adapt to a new staining protocol. Importantly, UPA is designed such that it allows for continuous recalibration as it automatically detects and adjusts to shifts observed in the prediction distribution over time. We showcase this by simulating multiple acquisition shift scenarios including the introduction of new scanners and updates to the image processing software. We discuss data requirements, assumptions, and limitations of the proposed method. We believe this work is of interest to anyone concerned about the safety and reliability of medical imaging AI including AI developers, healthcare professionals, patients, regulators, and policymakers.

## Results

### Datasets

We use four breast mammography datasets with side-wise biopsy-confirmed malignancy labels. For each participant, four images are recorded (two views per breast). The training dataset $D_{train}$ is from OPTIMAM[21] which is an enriched dataset from the UK, whereas all other evaluation sets are screening datasets from three sites in the UK and four sites in Hungary representative of the real-world populations in the respective national breast cancer screening programmes. For all evaluation sets, if there was more than one study per participant, we kept only one (randomly sampled). We additionally evaluate the effectiveness of UPA on WILDS Camelyon17, a publicly available histopathology dataset[22]. This dataset was designed to study dataset shifts caused by variations in staining protocols from one hospital to another. The dataset contains patches from whole-slide images with patch-wise labels indicating whether the tissue is cancerous or not. The training dataset is composed of data from three hospitals and the reference set is sampled from the same sites. Evaluation is on two unseen datasets from two new hospital sites. A summary of the dataset characteristics can be found in Tables 1 and 2.

### Experimental setup

We designed several realistic scenarios of acquisition shift to evaluate the effectiveness of UPA in the context of breast cancer detection in digital mammography and tissue classification in histopathology. UPA is a simple method aligning model predictions from an unseen acquisition domain (e.g., data acquired with a new scanner) to the reference prediction distribution recorded on a known domain (e.g., data from the validation set that was used during method development). Prediction alignment is achieved using piecewise linear cumulative distribution matching. For the two applications of mammography and histopathology, we train standard deep convolutional neural network models for image classification, calibrated on the validation data to yield an operating point where sensitivity equals specificity. More details about UPA, the AI models, and the settings for each scenario can be found in the Methods section.

### Scenario 1: Deployment to a new site

In this first set of experiments, we simulate the scenario where we want to deploy a model that has been optimised on validation data from a reference site to new unseen sites. Here, an acquisition shift may occur due to the use of different scanners at new sites (digital mammography), or more broadly due to differences in the imaging protocol

**Table 1 | Statistics of the breast cancer screening datasets**

| Dataset | Device | Country | Type | N cases | N images | % positives images |
|---|---|---|---|---|---|---|
| Scanner A | Hologic | UK | Reference set $D_{ref}$ | 3221 | 12,884 | 2.5% |
| Scanner B | IMS Giotto | Hungary | Unseen | 4152 | 16,608 | 1.8% |
| Scanner C | Siemens | UK | Unseen | 3904 | 15,616 | 1.9% |
| Scanner D | GE Healthcare | UK | Unseen | 4115 | 16,460 | 1.8% |

Source data are provided as a Source Data file.

**Table 2 | Statistics of the datasets for the WILDS Camelyon17 Histopathology dataset**

| Dataset | Type | N images | % positives |
|---|---|---|---|
| Sites S1–S3 | Reference set $D_{ref}$ | 33,560 | 50% |
| Site S4 | Unseen | 34,904 | 50% |
| Site S5 | Unseen | 85,054 | 50% |

Source data are provided as a Source Data file.

including the use of different staining protocols (histopathology). For the breast cancer detection task, we use a test dataset from a scanner that has not been seen during model development. For the histopathology tissue classification task, the evaluation data comes from another hospital using different staining protocols. In both cases, we assume that the corresponding classification models have been pre-calibrated on reference validation data (prior to deployment to new sites). The classification threshold has been selected at the clinical operating point where SEN equals SPC on this reference data. We first evaluate the performance of the model in terms of SEN/SPC on the new data prior to using UPA. Then we apply the proposed alignment technique to evaluate its effect on threshold shift. Metrics are reported in terms of image-wise area under the receiver-operating characteristic curve (ROC-AUC) and SEN/SPC, in Fig. 2 for breast cancer detection and in Fig. 3 for Camelyon17. Additionally, in Tables 3 and 4 we also report Youden's Index[23], a compound measure taking into account simultaneously sensitivity and specificity, before and after applying UPA. We find that without UPA there is a substantial change in SEN/SPC balance, across all tasks and evaluation datasets. Without UPA the model is no longer performing at the desired operating point. Conversely, results show that after applying our alignment method the SEN/SPC balance is largely restored for the unseen data. Additionally, in Supplementary Note 1, we show that UPA also works independently of the particular choice of operating point (e.g., predefined target specificity).

### Scenario 2: Transition to a new scanner

Here, we simulate a scenario requiring continuous model updates where a new scanner is installed, and an old scanner is gradually decommissioned. There is a planned transition period where the old and new scanners are used in parallel with a gradual switch to the new scanner. At the start, there is only data from scanner A, at the end there is only data coming in from scanner B. Classification thresholds of the AI model were optimised for scanner A. Results can be found in Fig. 4 (top), where the left-hand side depicts the number of cases processed by each scanner over time for each scenario and the right-hand side shows the corresponding results in terms of the model's sensitivity and specificity over time, with and without applying UPA overall and scanner-wise. From the figure, we observe that the introduction of the new scanner would lead to substantial drifts in SEN/SPC when not using UPA, whereas applying UPA preserves the desired SEN/SPC balance over time.

### Scenario 3: Addition of a new scanner

Similar to the scenario above, here a new scanner B is installed, however, the old scanner A will continue to be used in parallel. The new

scanner B is introduced gradually, meaning the total number of scans increases over time until there is an equal proportion from both scanners, resulting in the overall number of cases doubling. Classification thresholds were optimised for scanner A. Results can be found in Fig. 4 (bottom row), again UPA is able to preserve a balanced SEN/SPC over time.

### Scenario 4: Image processing update

In this scenario, we simulate the effect of an OEM update applied to the image processing algorithm for an existing scanner. The deployment site is assumed to operate a single scanner and at the time of AI deployment, the classification thresholds are optimised for the initial version of image processing software. At a later time point T1, we assume that the scanner manufacturer has applied a software update to their image processing algorithm resulting in an increase in image sharpness. Such updates are largely outside the control of the AI model developers. The results of this simulation can be found in Fig. 5. The increase in image sharpness induces a sudden and substantial change in the SEN/SPC balance. Applying UPA results in a rapid adaptation to the new image characteristics, with the model quickly performing at the desired SEN/SPC trade-off.

### Effect of dataset sizes

To gain more insights on the amount of data needed for obtaining good performance in the alignment phase of UPA, we analyse the sensitivity of the alignment method to the size of the reference set (from the source domain), as well as to the size of the alignment set (from the new domain). Results for mammography scanner B can be found in Fig. 6, with additional results for other datasets provided in Supplementary Note 2. These show that an alignment set as small as 250 cases (1000 images) is enough to recover the SEN/SPC shift across mammographic datasets. For the reference set, with 500 cases in the alignment set, we find that 1,000 cases are sufficient for stable results. In practice, one can use the same reference set as used for model validation and selection of the classification threshold.

## Discussion

We have proposed an effective method to correct clinically relevant performance drift attributable to changes in the image acquisition pipeline (e.g., replacement of scanners, updates of image processing software, use of different staining protocols). This has been demonstrated across various scenarios from one-off changes when deploying to a new site to gradual and progressive drifts in acquisition characteristics over time. Our experiments show that a generalisable ROC-AUC does not necessarily mean that selected classification thresholds (i.e., clinical operating points) generalise to unseen data. As shown in Figs. 2 and 3, the studied models did generalise in terms of ROC-AUC across all datasets while the SEN/SPC balance varied substantially across sites when applying a predefined classification threshold. Our results demonstrate that the proposed unsupervised prediction alignment method can successfully mitigate threshold shifts on unseen datasets in two very different image classification tasks. UPA was able to recover the desired SEN/SPC trade-off, without using any labels from the unseen target domain. Importantly, this simple yet effective method can be used for continuous model updates and helps

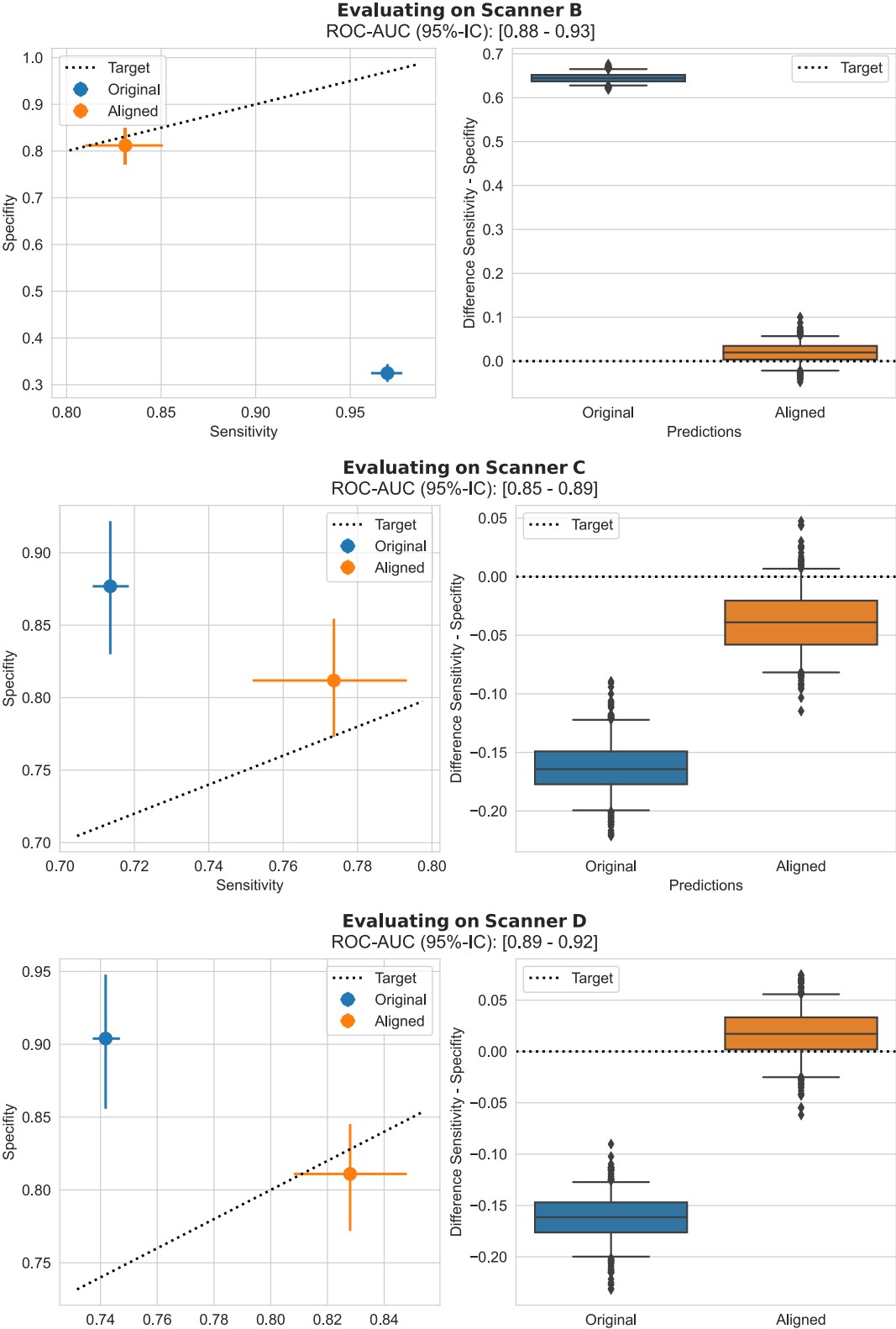

**Fig. 2 | Scenario 1: Deployment to a new site—breast screening task.** Left column: Specificity in function of sensitivity before and after prediction alignment. For this analysis, we sample an evaluation set (of 2500 cases) and a disjoint alignment set (of 1000 cases) from all available cases, this sampling is repeated 500 times with replacement. Sensitivity, specificity, ROC-AUC are measured over these 500 bootstrap samples and results are reported in terms of average results over the bootstrap samples and error bars depict the 95%-bootstrap confidence interval for each metric. Right column: the difference between sensitivity and specificity before and after alignment. Boxplots are constructed from 500 repeated sampling of evaluation and alignment sets; each box shows the 25%, 50% and 75% percentiles of the bootstrap distribution; whiskers denote the 5% and 95% percentiles and any point outside of this range is represented as an outlier. UPA is effective at recovering the desired sensitivity/specificity balance across all out-of-distribution datasets. Source data are provided as a Source Data file.

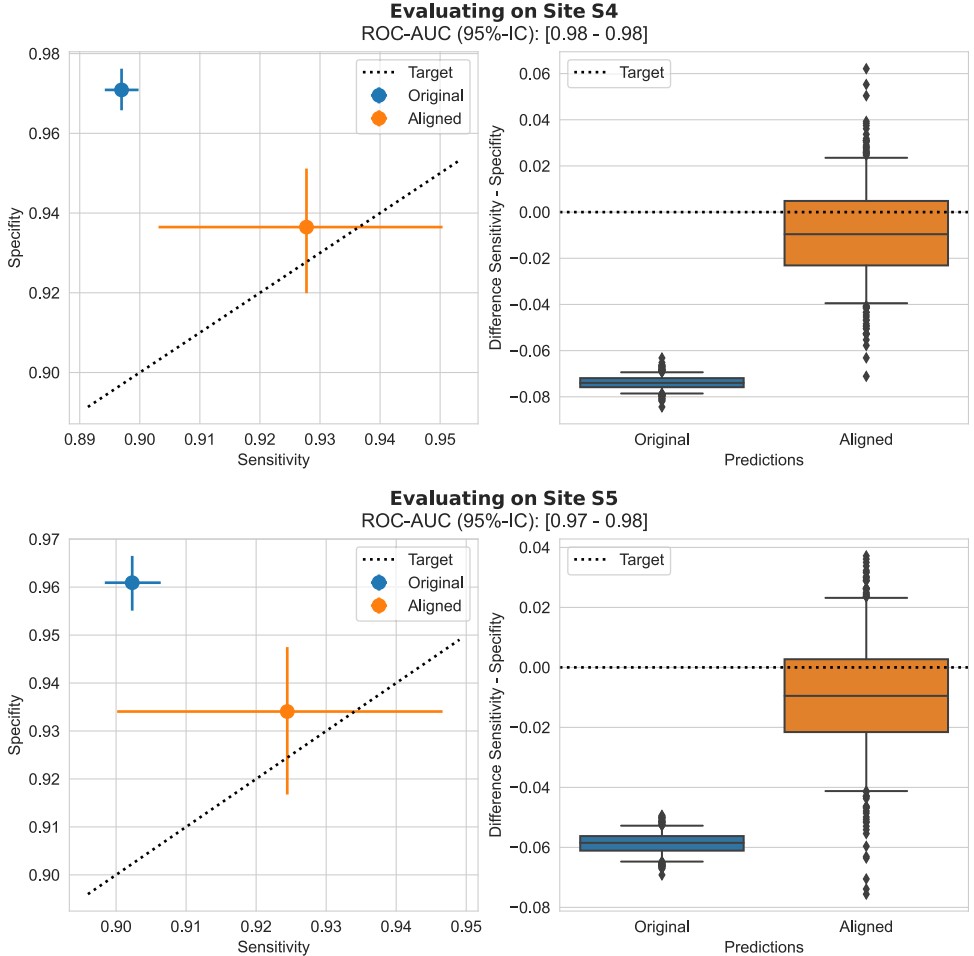

**Fig. 3 | Scenario 1: deployment to a new site–WILDS Camelyon17 histopathology task.** Left column: Specificity in the function of sensitivity before and after prediction alignment. For this analysis, we sample an evaluation set (of 15,000 patches) and a disjoint alignment set (of 5000 patches) from all available patches, this sampling is repeated 500 times with replacement. Sensitivity, specificity, ROC-AUC are measured over 500 repeated sampling of evaluation and alignment sets and results are reported in terms of average results over the bootstrap samples and error bars depict the 95%-bootstrap confidence interval for each metric. Right column: the difference between sensitivity and specificity before and after alignment. Boxplots are constructed from 500 bootstrap samples of evaluation and alignment sets; each box shows the 25%, 50% and 75% percentiles of the bootstrap distribution; whiskers denote the 5% and 95% percentiles and any point outside of this range is represented as an outlier. UPA is effective at recovering the desired sensitivity/specificity balance across all out-of-distribution datasets. Source data are provided as a Source Data file.

in the case of progressive performance drift. Our sensitivity analysis shows that UPA requires only limited amounts of data from the shifted distribution and can re-use existing validation data as the reference set. With only $N = 500$ cases from the target domain and $N = 1000$ cases from the source domain, we observed near-optimal and stable results for recalibration of the model outputs. The fact that only a limited amount of data from the target domain is required is important for the practical implementation of UPA. Used as a continuous performance monitoring and recalibration tool, UPA could run over a relatively small time window of new incoming data. In breast cancer screening,

for example, acquiring a set of 500 new cases is often only a matter of days at many screening sites.

In practical use-cases where several scanners or acquisition protocols are used at the same site (such as scenarios 2 and 3 above), it is worth noting that UPA is applied in a scanner/protocol-wise manner. That is, we fit one alignment algorithm for each scanner separately based on the most recent set of patient scans acquired with this specific scanner. This allows them to operate fully independently. This fine-grained device-wise adaptation is particularly advantageous in cases where the proportion of scans processed by a given type of scanner varies on a day-to-day basis or one of them is suddenly not operating for maintenance. Similarly, if one scanner has a software update affecting the model predictions, only this scanner would have to automatically recalibrate without affecting other scanners.

Compared to approaches that directly adjust the classification threshold instead of the model predictions[24], UPA has the advantage of maintaining relative calibration of the model as it preserves the shape of the distribution between reference and target domain. This enables more informed decisions based on the value of the prediction associated with a given sample (e.g., to assess the prediction uncertainty). Conversely, if one simply adapts the threshold, there is no guarantee

**Table 3 | Youden's Index for scenario 1, deployment to a new site, breast screening**

| Dataset | Before correction | After correction |
| --- | --- | --- |
| Scanner B | 0.295 (0.011) | 0.651 (0.021) |
| Scanner C | 0.599 (0.023) | 0.594 (0.020) |
| Scanner D | 0.644 (0.024) | 0.640 (0.021) |

Reported as average over 500 bootstrap samples, with standard deviation in brackets. Source data are provided as a Source Data file.

that the shape of the prediction distribution is preserved, which may render it impossible to interpret the absolute value of the prediction. This is important, as it has been shown that model calibration is not necessarily transferable when moving from validation to unseen test domains[25]. In Fig. 1, we saw how the shape of the distribution changes before and after alignment and how the alignment preserves the shape of the reference. When comparing the expected calibration error (ECE)[26], we found that after alignment the ECE is preserved between the reference (ECE = 0.13) and the aligned distribution (ECE = 0.14), whereas it was twice as large on the original predictions before alignment (ECE = 0.29). The preservation of the shape of prediction distribution may be of particular importance in applications utilising the absolute value of the model predictions such as uncertainty-aware triaging[1,2], or when used in risk prediction models[27].

In terms of limitations, it is important to highlight that UPA focuses on addressing the effects of acquisition shift only, it is not designed to tackle other types of distribution shifts that also may cause performance drift such as population, prevalence or annotation shift[28,29]. Crucially, in UPA we assume that the prevalence is (roughly) preserved between reference and target domains. This is a reasonable assumption in many practical clinical scenarios, e.g., in screening programs where the expected prevalence is known and usually similar across sites with similar populations. Note that if the prevalence

assumption is violated, but the target prevalence is known, or it is possible to gather labels on the unseen dataset, UPA can still be applied by ensuring that the prevalence in the chosen reference set matches the target prevalence (e.g., using re-sampling techniques). However, if no or only insufficient information is available about the causes of the distribution shift between the reference set and the new target domain, UPA should not be used for automatic recalibration without further investigation. In practice, we would envision UPA to be used alongside comprehensive monitoring of the input data including meta information about the patient population. This would enable the detection and flagging of unexpected changes in the patient population. Monitoring of base demographics is already standard practice in screening programmes. This could be complemented with automated monitoring tools, for example using auxiliary AI models, to predict patient and data characteristics from the input images (e.g., age, breast density, image quality, etc.) and methods that have been specifically developed for drift detection[19]. The detection of shifts is an important aspect of continuous performance monitoring. Note that UPA does not have to be applied in real-time, meaning for deployment it may be reasonable to implement a time delay (say a few days) between the detection and the correction of the distribution shift. This would facilitate a human-in-the-loop inspection of whether other sources may have contributed to the detected shift. Here, we note that mixing effects of different sources of potential bias in the datasets can complicate the root cause analysis of data distribution shift[29]. Still, even in such settings, UPA can be employed for the detection of distribution shift during deployment, as we have illustrated in the case of prevalence shift in Supplementary Note 4.

Secondly, we should note that UPA is designed to tackle threshold shift, not model generalisation overall. Importantly, we assume here that the acquisition shift primarily induces a shift in the model predictions invalidating the selected operating point while preserving ROC-AUC across domains. All models investigated in this study satisfy

**Table 4 | Youden's Index for scenario 1, deployment to a new site, histopathology**

| Dataset | Before correction | After correction |
|---------|-------------------|------------------|
| Site S4 | 0.868 (0.003) | 0.864 (0.005) |
| Site S5 | 0.859 (0.006) | 0.863 (0.004) |

Reported as average over 500 bootstrap samples, with standard deviation in brackets. Source data are provided as a Source Data file.

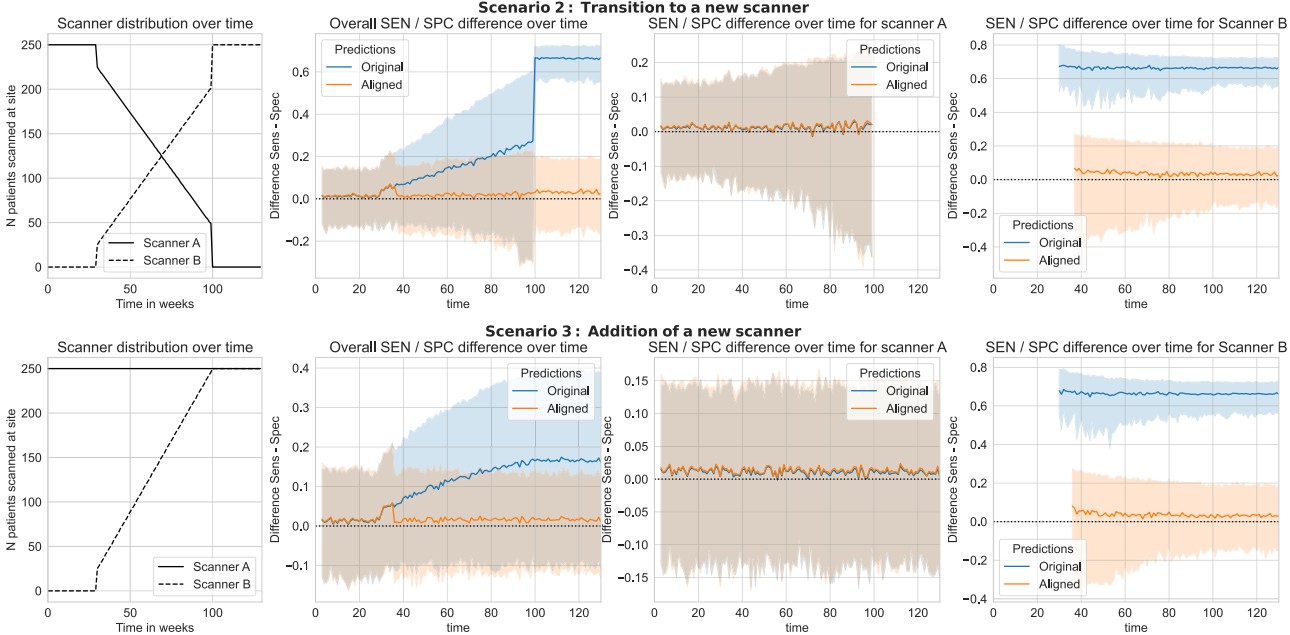

**Fig. 4 | Scenarios 2 and 3: Prediction alignment over time under continuous acquisition shift.** Each simulation is repeated 250 times, solid lines depict the average difference between sensitivity and specificity across all bootstrap samples and shaded regions denote the 5%-95% percentile bootstrap confidence interval. Plots in the left column depict the number of scans processed by scanner A and scanner B over time for each scenario. Plots in the middle column compare the evolution of the sensitivity-specificity balance over time with and without applying UPA across all scans. The right-most two columns compare the evolution of the SEN/SPC balance scanner-wise. The goal is to avoid a drift between sensitivity and specificity in the presence of a gradual acquisition shift. The proposed method successfully maintains a null difference between sensitivity and specificity over time, whereas the non-adapted model can lead to dramatic shifts in the sensitivity-specificity balance. Source data are provided as a Source Data file.

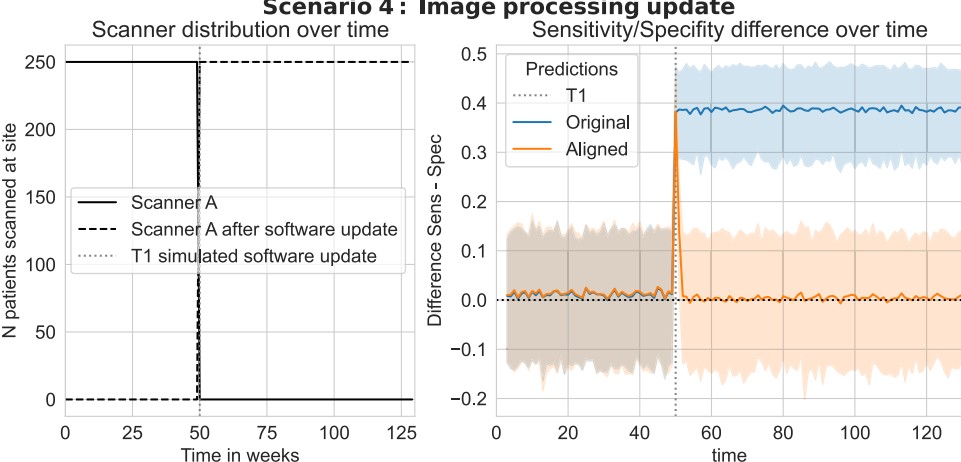

**Fig. 5 | Scenario 4: Prediction alignment over time in the case of software update.** Each simulation is repeated 250 times, solid lines depict the average difference between sensitivity and specificity across all bootstrap samples and shaded regions denote the 5%–95% percentile bootstrap confidence interval. Plots in the left column depict the number of scans processed by scanner A and scanner B over time for each scenario. Plots in the right column compare the evolution of the sensitivity-specificity balance over time with and without applying UPA. The goal is to avoid a drift between sensitivity and specificity in the presence of a gradual acquisition shift. The proposed method successfully maintains a null SEN/SPC difference over time, whereas the non-adapted model can lead to dramatic shifts in the sensitivity-specificity balance. Source data are provided as a Source Data file.

this requirement and generalise well in terms of ROC-AUC, i.e., their ability to separate classes is largely preserved even when the input image characteristics change. As we apply piecewise linear interpolation to the model outputs, UPA guarantees that the ROC-AUC is identical before and after the alignment (as it does not change the relative order of the predictions). As such, UPA's ability to recover the expected SEN/SPC balance is not dependent on the classification performance, as long as the ROC-AUC is preserved across domains. If the ROC-AUC of a model were to drop significantly from one domain to another, UPA is expected to perform suboptimally, and will not be able to recover ROC-AUC performance on unseen data. Tackling issues related to performance drop in terms of overall class separability, i.e., measured by a drop in ROC-AUC, is the focus of a large body of works in domain adaptation (DA)[30,31] and domain generalisation (DG)[32,33]. These lines of work are complementary as DA/DG methods focus on building generalisable models in terms of ROC-AUC, with usually little attention on threshold shift. Here, UPA addresses an unmet need by focusing on the generalisability of model calibration and its pre-defined, clinical operating points. The proposed method is by no means meant as a replacement for traditional DA/DG techniques but should be considered in addition. While using DA/DG methods during model development may result in increased robustness to acquisition shift, there is no guarantee of the performance in new unseen domains, in particular in terms of clinical metrics. Here, UPA adds an important safeguard for identifying performance drift in new deployments. Additionally, we believe that local validation on representative data[5], regular AI audits[34], and a careful analysis of discordant cases remain critical components for safe clinical deployment and the assurance that AI continues to be safe over time.

Additionally, where we propose UPA in a medical imaging use case, global medical device regulations must be considered. Strict control of a cleared medical device is required to ensure continued safety and effectiveness following release to a market, especially when considering an unsupervised modification to parameters. As such, we do not claim that applying UPA would dispense users of standard auditing procedures involving comparison to human assessment. However, integrated in the workflow it can allow for automatic correction to happen before the next auditing round, as data acquisition updates may happen at any time. Modern AI software as a medical device regulations include the option of 'change protocols' to allow AI devices to improve over time with respect to clinically relevant metrics

but must be negotiated and agreed upon with the regulators and include appropriate human oversight in deployment[35]. Thorough validation of processes influencing the performance of a medical device, as well as comprehensive post-market surveillance and continuous model monitoring are critical to the success of this approach in a regulated environment.

## Methods

### Unsupervised prediction alignment

UPA consists of applying linear piecewise cumulative distribution matching between the prediction distribution on the unseen dataset and the reference prediction distribution. Piecewise linear matching is also known as 'histogram matching' and is a well-known technique in the field of image processing where it is traditionally applied to standardise image intensities[36]. In our setting, this matching algorithm is applied to the model's output predictions. To fit the matching algorithm, we require a set of predictions from the reference domain (the 'reference set'), as well as a set of predictions coming from the out-of-distribution domain, the 'alignment set'.

The matching algorithm then consists of the following three simple steps:

- Compute the observed cumulative distribution of the predictions on the reference and alignment sets.
- Fit a linear interpolator to match the alignment set cumulative distribution to the reference set cumulative distribution.
- Apply this linear interpolator to any new test prediction coming from the unseen domain.

Pseudo-code for the matching algorithm is provided in Supplementary Note 3, and a fully functional Python implementation is provided in the Supplementary Code.

### AI model

For the purpose of this study, we trained a simple ResNet-50[37] convolutional neural network on $D_{train}$ (see above) with a cross-entropy loss for each task. For breast cancer applications we apply breast masking and intensity normalisation before processing the images with the AI model. The model is trained on an enriched breast screening dataset with 12,285 cases (22.7% positive images). For the histopathology model, we train on the official training split from the Camelyon17 dataset with 302,436 images (50% positives). After

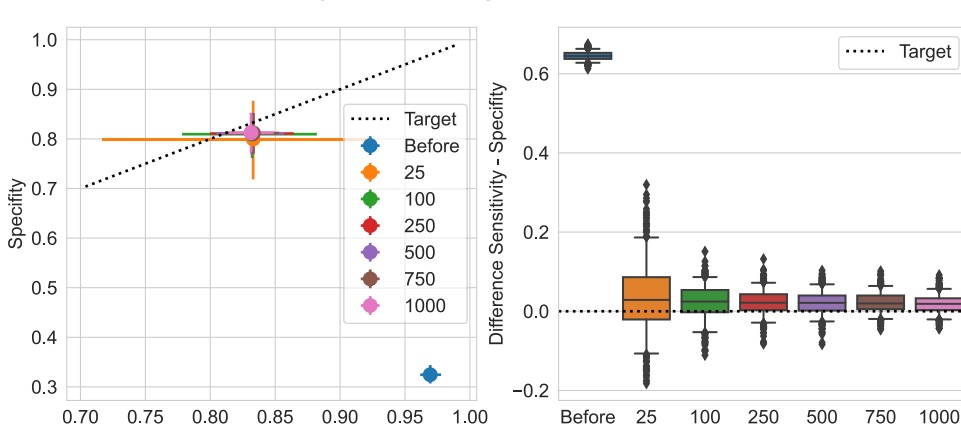

Evaluating the effect of alignment set size on Scanner B

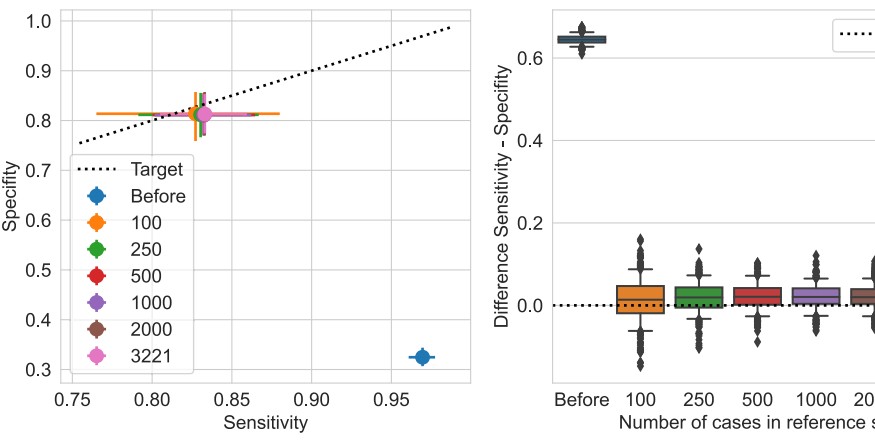

Evaluating the effect of reference set size on Scanner B

**Fig. 6 | Sensitivity analysis on the effect of sizes of alignment and reference sets on the performance of the alignment method (scanner B).** In the sensitivity analysis on the size of the alignment set (top), we used the full reference set (3221 cases). Results are reported over 500 bootstrap samples. For the alignment size analysis, each bootstrap sample is created by sampling one alignment set of the size of interest from all available cases as well as one evaluation set (*n* = 2500 cases). For the reference size analysis, each bootstrap sample is created by sampling one reference set of the size of interest from all available cases as well as one evaluation set (*n* = 2500 cases). On the left, the points depict the average SEN/SPC over samples and error bars represent the 95% bootstrap confidence interval. On the right,

each box shows the 25%, 50% and 75% percentiles of the bootstrap distribution; whiskers denote the 5% and 95% percentiles and any point outside of this range is represented as an outlier. This analysis shows that with as few as 250 cases in the alignment set, we already get very good results, stable across repeated sampling experiments. In the analysis below, we measured the performance of methods for different reference set sizes, we used 500 cases for alignment and varied the size of the reference set. We can see that the method is not too sensitive to the size of the reference, even if the more data the reference distribution comprises, the better the results get. Source data are provided as a Source Data file.

training, we select the classification threshold at the balanced operating point (where sensitivity equals specificity) on the in-distribution validation set $D_{ref}$, which we denote by the reference set (see Tables 1 and 2). Note that the trained models generalise well to their out-of-distribution settings in terms of ROC-AUC but not in terms of sensitivity/specificity (as shown in the Results section).

**Details for 'Scenario 1: Deployment to a new site'**
We use the AI model along with a preset classification threshold optimised on $D_{Ref}$. We first measure the performance of this model when deployed to a site by sampling an evaluation set of 2500 cases from the out-of-distribution dataset (resp. 15,000 patches for the histopathology application) and evaluate the performance of the model in terms of sensitivity and specificity on this set. Additionally, we sample a disjoint set of N = 1000 cases (resp. 5000 patches) to fit the prediction alignment algorithm. For the mammography application, we use the full reference set $D_{ref}$ as our reference distribution (3221 cases). For the histopathology task, we sample 5000 images from $D_{ref}$ to use as our reference distribution. We then re-evaluate the

performance of the model on the evaluation set after alignment to measure the effectiveness of the proposed method to reduce the sensitivity-specificity drift. Sampling of the alignment and evaluation set is repeated 500 times to obtain uncertainty estimates for the reported metrics. Note that, in the breast screening dataset, for each case, we have 4 images per study.

**Details for 'Scenarios 2 to 4: Continuous model updates'**
In these scenarios, we wish to study the effectiveness of our method for continuous model updates in the presence of progressive temporal acquisition shift for the mammography task. To this end, we simulate a data flow by defining a function mapping time to a given scanner distribution describing each of the above scenarios. To get the reference set and the simulation sets for scanner A, we split the original $D_{ref}$ into two disjoint subsets: one randomly sampled subset of 1200 cases is used as the reference set (to set the starting classification threshold and as the target of the alignment throughout the simulation); the remaining split of 2221 cases is then used as a data source for the simulation itself. We assume here that the time

unit in these simulations is in weeks. For each time point T, we sample data according to the expected scanner distribution at point T−as defined by each scenario. At any given time point T, we use a running window composed of the data of the last two weeks T-2 and T-1 for fitting both alignment algorithms (separately for each scanner). We then evaluate the performance of the model before and after alignment on the set sampled for time T both scanner-wise and overall. We additionally ensure that−at any given time point T−the running window set (samples of T-1 and T-2) is disjoint for the evaluation set (samples of T). In scenarios 2 and 4 we assume that the total number of cases per time point (week) is constant at $N = 250$ cases. In scenario 3, we start with $N = 250$ at the beginning of the simulation and progressively increase the number of cases screened as the second scanner ramps up until we reach $N = 500$. In scenario 4, we simulated the software update by increasing the sharpness of the input images, we used Scanner A images in this scenario too.

### Ethical information

The internal breast cancer datasets from the UK and Hungary were collected previously with ethical approval from the UK National Health Service (NHS) Health Research Authority (HRA) (Reference: 19/HRA/0376) and the Medical Research Council, Scientific and Research Ethics Committee in Hungary (ETT-TUKEB) (Reference: OGYÉI/46651–4/2020). The original study was performed in accordance with the principles outlined in the Declaration of Helsinki for all human experimental investigations. The need for informed consent to participate was reviewed by HRA and ETT-TUKEB and confirmed to not be required as the study involved the secondary use of retrospective and pseudonymised data. The present study made secondary use of a fully anonymised version of this previously collected data. For the OPTIMAM breast cancer dataset, the OPTIMAM project obtained renewed HRA approval and a favourable ethical opinion in July 2019 (Reference: 19/SC/0284) for a renewable period of 5 years, to collect images and data from participating sites for the creation of a research database and to add new collection sites. Patient consent was waived by the NHS HRA South Central−Oxford C Research Ethics Committee under the NHS constitution which clarifies that the collection of de-identified data without patient consent is permissible. The present study used a fully anonymised version of the OPTIMAM dataset. The use of the histopathology datasets is exempt from ethical approval as the analysis is based on fully anonymised, secondary data which is publicly available.

### Reporting summary

Further information on research design is available in the Nature Portfolio Reporting Summary linked to this article.

## Data availability

All anonymised model outputs supporting the findings described in this manuscript are publicly available in our code repository at https://github.com/biomedia-mira/upa. Source data are provided in this paper which includes the data used to plot the graphs shown in the figures, as well as tables. Access to the OPTIMAM breast cancer dataset can be requested on the project's website: https://medphys.royalsurrey.nhs.uk/omidb/. The WILDS-Camelyon dataset is publicly available under a Creative Commons CC0 license as part of the WILDS benchmark and is readily available for download to anyone. Downloading instructions can be found at https://wilds.stanford.edu/get_started/. Official data splits were used as part of this study. Raw images from the internal breast cancer imaging datasets from the UK and Hungary were obtained under commercial licences and are not publicly available. Requests for further information can be made via email to the corresponding authors and will be processed within four weeks. Source data are provided in this paper.

## Code availability

The implementation of UPA together with code to reproduce all results and figures are provided in the Supplementary Code and are also made publicly available on https://github.com/biomedia-mira/upa. The code is written Python 3.10 along with the following packages: seaborn-0.12.2, matplotlib-3.7.1, numpy-1.25, pandas-2.0.2, scikit_learn-1.2.2, jupyter-1.0.0, tqdm-4.65.0, notebook-6.5.4.

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

## Acknowledgements

B.G. is grateful for the support from the Royal Academy of Engineering as part of his Kheiron Medical Technologies/RAEng Research Chair in Safe Deployment of Medical Imaging AI.

## Author contributions

M.R., G.K. and B.G. conceptualised and designed the study. M.R. implemented the algorithms, conducted the experiments, and performed the statistical analysis. M.R., G.K. and J.Y. performed data curation and pre-processing. M.R., G.K., J.Y., B.G. and T.R. interpreted the results and verified the underlying data. N.S., J.J.J. and E.A. collected the data and provided medical expertise. T.R. and P.K. provided administrative support and project supervision. A.H. provided regulatory oversight. M.R., G.K. and B.G. wrote the initial manuscript. All authors edited and reviewed the manuscript and approved the final version.

## Competing interests

G.K., J.Y., A.H., P.K., T.R., B.G. are employees of Kheiron Medical Technologies with stock options as part of the standard compensation package. M.R. was a paid intern of Kheiron at the time of this work. All other authors declare no competing interests.
