## [Peer Review File · Nature Communications]

Automatic correction of performance drift under acquisition shift in medical image classificationREVIEWER COMMENTS

Reviewer #1 (Remarks to the Author):

This paper explores the impact of different types of drift on medical image data analysis. It proposes a method, unsupervised prediction alignment (UPA) for handling different types of acquisition drift (e.g. where a scanner is replaced / software is updated). No annotation or population drift explored.

The problem is important and timely. The difference in distributions from the Mammogram scanners in Fig 1 highlight this. There can be many different causes of drift that will impact the analysis of medical image data and it is important to both identify this and handle it.

The method in this paper, UPA, involves computing the observed cumulative distribution of the predictions on the reference and alignment sets, fitting a linear interpolator to match the alignment set cumulative distribution to the reference set cumulative distribution, and applying this this linear interpolator to any new test prediction coming from the unseen test data.

Results are convincing with alignment restoring the balance of sens / spec. Fig 4 is very convincing, showing how the alignment ensures no drift occurs throughout beyond a small positive bias in sens-spec (could this be fine tuned to ensure zero bias?). Figure 5 shows almost instant "snap-back" to the correct performance.

For these experiments, sens and spec were considered equally important. It may be interesting to see other scenarios where sens or spec takes precedence over the other (e.g. screening). I would at least like to see some more discussion of this (but some additional results would also strengthen the conclusions of the UPA approach).

I wonder what would happen if the distributions in the reference and the new set are more substantially different (e.g. different shapes)? Some discussion (preliminary indicators) would also improve the paper.

Only one CNN model explored here - I guess that it is likely that the choice of AI model would have a considerable impact on these results (for example, do better performing / generalising models update more easily)? This is probably beyond the scope of the paper but would be worth discussing.

Reviewer #2 (Remarks to the Author):

The authors present a method to adjust for one drift scenario for prediction models targeting binary classification tasks. The scenario is when the model used has good generalization capacity, essentially with retained AUC-ROC, but the shift affects the best choice of operating point. The method proposed consists of remapping the model prediction scale (for which a threshold makes the binary cut-off) such that the distribution of the output for the target domain data becomes the same as for the original validation data.

The strengths of the work include that it targets a highly relevant clinical use case and suggests an effective method for addressing it, feasible also considering actual clinical prerequisites. The fact that no annotations/labels on the target data is needed is a major benefit. The experiments nicely show that the intended effect is achieved. Moreover, the limitations of the work are very clearly discussed, mainly that the targeted scenario only covers one part of the domain shift challenges.

The method employed is straightforward. It is so straightforward that I would be surprised if it is not already in use, for instance in calibration procedures for commercial AI products. However, it appears that this specific approach has not been published previously (I haven't found any such references either), and we shouldn't punish good solutions because they are simple - on the contrary.

I would suggest to change the title to reflect the narrow scenario targeted. The use case targeted is the "threshold shift for well generalizing methods" described, whereas the title's wording "performance drift" can be interpreted as covering a much wider range of domain shift challenges.

I also think that the paper would be improved if the interplay with other shifts was investigated in some way. I agree with the authors that the proposed method probably could be used as an aid in detecting other shifts, and an experiment showing how it behaves in such a scenario would be enlightening. (If there are space issues, I'd suggest moving fig 4 and/or 5 to supplementary, nice experiments but unsurprising results given fig 2 & 3.)

A detail is that I'd be interested in also seeing the MCC measure as a complement to sensitivity and specificity in the experiments. As sensitivity and specificity have limitations in imbalanced settings, MCC is a good sanity check that there are no unexpected effects.

A minor detail is that in line 307, sens=spec is referred to as an optimal operating point which is not really true, the optimal choice depends on the use case as the authors correctly say in the introduction.

To summarize, the paper is well written and has nice contributions. It could be argued that the amount of scientific contribution is a bit limited for Nature Communications, but for me some deepened investigation such as what's proposed above would take it over the bar.

Reviewer #3 (Remarks to the Author):

The authors proposed an automatic recalibration method, unsupervised prediction alignment (UPA), that requires no ground truth annotations and only limited amounts of unlabelled example images from the shifted data distribution. The method is intended to correct for performance drift in AI software caused by changes in data acquisition such as the replacement of scanner hardware or updates to the image processing software. They conducted simulation experiments to demonstrate that UPA can restore the expected performance under several scenarios of image acquisition shift.

The UPA method was relatively simple and could be automated. The authors expected it to be applicable if the data distribution shift was caused by acquisition shift while the population and prevalence remained stable. A major issue is that since the method was applicable only to data acquisition shift and might cause unwanted calibration errors if the data distribution shift was caused by other types of shifts, it is unclear how the method could avoid being mistakenly applied to other types of shifts if it was implemented as an automated procedure in the clinical workflow.

Another major issue is that the authors assumed that the data distribution shift due to, e.g., a scanner change, could be corrected by a simple "piecewise linear cumulative distribution matching", which may not be valid in general. For example, mammography systems from different vendors used different target/filter, imaging techniques, detector, and post-processing techniques. An AI software trained with images from one type of scanners cannot be simply applied to another type, or the ROC curve and AUC will change even if the patient population and disease prevalent remain the same. Furthermore, even with the same scanner, acquisition drift could degrade the image quality and cause a degradation in the ROC curve and AUC of the AI software. No simple matching of the AI output distribution can bring the sensitivity and specificity to the same levels as the reference. As such, the applicability of the proposed UPA will be limited to very few special situations. The use a relatively broad term to describe a very narrow situation is misleading although it was mentioned at the end (Discussion: "acquisition shift primarily induces a shift in the model predictions invalidating the selected operating point...").

specific comments:

1. Introduction, the authors correctly pointed out that “The topic of performance monitoring is a pressing matter for the practical deployment of machine learning models”. However, it is unclear how the proposed method could automatically detect performance drift. The simulated scenarios in their study assumed that a new/different scanner would be introduced or an old one would be replaced, or a software upgrade would be performed, etc. With such known events, the UPA could be invoked to make corrections, if appropriate. In real-world clinical practice, it is difficult to detect any shift in the AI output data distribution unless a QA monitoring procedure is implemented to collect AI performance data with or without respect to some “reference truth” statistically over time.

2. As commented above, it is unclear how the proposed method could automatically detect model performance shift and determine whether the shift was within those the UPA could handle. In general, many factors can cause shifts in the image quality and image characteristics and the different types of shifts can mix together. The statement “there is an unmet need for methods allowing to automatically detect and correct model performance drift in the absence of any diagnostic information ... This is precisely the focus of this work.” appears to imply that the proposed method and study could meet this need. Please clarify.

3. Datasets, “all other evaluation sets are screening datasets from three sites in the UK and four sites in Hungary representative of the real-world populations in the respective national breast cancer screening programmes.”: Please specify how the seven sites were grouped into the four datasets described as scanners A-D in Table 1, and whether each dataset actually were collected with a single scanner type, and if so, what type for each set. As described, the seven sites came from two national screening programs from two countries, how did the authors determine that data distribution shift were caused by scanner difference or software upgrade only? One would expect that population shift, for example, could be a major factor between datasets from the two countries. It is important to be specific about the characteristics of these datasets because they were used as examples to demonstrate the proposed method could work well. The results of the experiments gave the impression that the method was quite versatile and applicable to international multi-site data from four different types of scanners.

4. Discussion, last paragraph “Additionally, ...”: The discussion here is general and important for deployment of medical AI to clinical use; however, it is not specific to the study or method proposed in this paper. The general justifications have been discussed in the Introduction. Concluding the study with this discussion is too broad and out of scope.

Response to reviewers

Many thanks to the reviewers for taking the time to assess our manuscript. We appreciate the positive feedback and valuable suggestions. We have incorporated the suggestions into the revised version of our manuscript. Please find our point-by-point responses below. In the revised version, all changes are highlighted with blue font.

Reviewer 1

Comment 1.1: This paper explores the impact of different types of drift on medical image data analysis. It proposes a method, unsupervised prediction alignment (UPA) for handling different types of acquisition drift (e.g. where a scanner is replaced / software is updated). No annotation or population drift explored.

The problem is important and timely. The difference in distributions from the Mammogram scanners in Fig 1 highlight this. There can be many different causes of drift that will impact the analysis of medical image data and it is important to both identify this and handle it.

Response: We thank the reviewer for the positive feedback and for recognising the value of our work.

Comment 1.2: The method in this paper, UPA, involves computing the observed cumulative distribution of the predictions on the reference and alignment sets, fitting a linear interpolator to match the alignment set cumulative distribution to the reference set cumulative distribution, and applying this this linear interpolator to any new test prediction coming from the unseen test data.

Results are convincing with alignment restoring the balance of sens / spec. Fig 4 is very convincing, showing how the alignment ensures no drift occurs throughout beyond a small positive bias in sens-spec (could this be fine tuned to ensure zero bias?). Figure 5 shows almost instant "snap-back" to the correct performance.

For these experiments, sens and spec were considered equally important. It may be interesting to see other scenarios where sens or spec takes precedence over the other (e.g. screening). I would at least like to see some more discussion of this (but some additional results would also strengthen the conclusions of the UPA approach).

Response: As pointed out by the reviewer in our experiment we chose an operating point where sensitivity and specificity are equally important. This is a design choice that we made for this study but is not a requirement for our method. One could choose any suitable operating point depending on the use-case requirements. Indeed, our method matches the prediction distributions between reference and target domains independently of the threshold choice.

To further demonstrate this in a concrete example, below we repeat our first experiment "deployment to a new site" with another operating point. In this additional experiment, we

chose an operating point on the reference domain such that “specificity = 90%”, our goal then is to maintain this 90%-sensitivity operating on the target domain after applying UPA.

Supp. Fig. **Scenario 1: Deployment to a new site - breast screening task - operating point “specificity 90%”**. Specificity in function of sensitivity before and after prediction alignment. Sensitivity, specificity, ROC-AUC are measured over 500 repeated sampling of evaluation and alignment sets and results are reported in terms of average results over the bootstrap samples and error bars depict the 95%-bootstrap confidence interval for each metric. UPA is effective at recovering the desired specificity across all out-of-distribution datasets.

We have updated the manuscript accordingly by adding these results to the supp. material (Note 1) and referenced them in the Results section, Scenario 1 subsection”.

Comment 1.3: I wonder what would happen if the distributions in the reference and the new set are more substantially different (e.g. different shapes)? Some discussion (preliminary indicators) would also improve the paper.

Response: Because we assume that (i) ROC-AUC is preserved across devices and (ii) prevalence is preserved across domains, we do not expect to see much more drastic changes in distribution shapes than the ones shown in Figure 1. Indeed, if more drastic changes were observed, this would likely suggest a change in ROC-AUC or be the consequence of a change in prevalence. Such shape changes could in fact be used for additional monitoring, to flag scenarios where a more in-depth analysis of the distribution changes is required. To illustrate this further, we have now added an additional experiment on the detection of drifts caused by prevalence shift. Please also see our response to Comment 2.3 and the corresponding changes in the revised manuscript.

Comment 1.4: Only one CNN model explored here - I guess that it is likely that the choice of AI model would have a considerable impact on these results (for example, do better performing / generalising models update more easily)? This is probably beyond the scope of the paper but would be worth discussing.

Response: As pointed out by the reviewer, in order to produce satisfying results UPA assumes the model to be generalisable in terms of ROC-AUC i.e. similar ROC-AUC across domains, this being one of the two main assumptions of our work. However, since UPA preserves the ranking of the predictions (i.e. ROC-AUC) across domains and focuses on matching prediction distributions, we do not expect UPA’s performance to be dependent on the underlying classification model performance. For example, in our breast screening

experiments the ROC-AUC associated with the classification models is ~ 0.89 for all scanners whereas on WILDS-Camelyon the ROC-AUC of the models is significantly higher $\sim .98$. Nonetheless, UPA works equally well at recovering the expected SEN/SPC for both tasks, the performance of the model simply impacts the reference value of SEN/SPC, not the ability to recover the balance when transferring to a new domain.

We have updated the last paragraph of the discussion to reflect these points and thank the reviewer for their suggestion.

Reviewer 2

Comment 2.1: The authors present a method to adjust for one drift scenario for prediction models targeting binary classification tasks. The scenario is when the model used has good generalization capacity, essentially with retained AUC-ROC, but the shift affects the best choice of operating point. The method proposed consists of remapping the model prediction scale (for which a threshold makes the binary cut-off) such that the distribution of the output for the target domain data becomes the same as for the original validation data.

The strengths of the work include that it targets a highly relevant clinical use case and suggests an effective method for addressing it, feasible also considering actual clinical prerequisites. The fact that no annotations/labels on the target data is needed is a major benefit. The experiments nicely show that the intended effect is achieved. Moreover, the limitations of the work are very clearly discussed, mainly that the targeted scenario only covers one part of the domain shift challenges.

The method employed is straightforward. It is so straightforward that I would be surprised if it is not already in use, for instance in calibration procedures for commercial AI products. However, it appears that this specific approach has not been published previously (I haven't found any such references either), and we shouldn't punish good solutions because they are simple - on the contrary.

Response: We thank the reviewer for recognising the value of our work. We appreciate the comment regarding the practicality of the method which we would hope would indeed make it very attractive to be incorporated into commercial solutions. We are not aware that similar techniques are currently in use, and our work may hopefully be of interest for AI developers.

Comment 2.2: I would suggest to change the title to reflect the narrow scenario targeted. The use case targeted is the "threshold shift for well generalizing methods" described, whereas the title's wording "performance drift" can be interpreted as covering a much wider range of domain shift challenges.

Response: Thanks for the suggestion. We have changed the title accordingly to reflect that we are specifically focusing on performance drift caused by acquisition shift. The new suggested title is: "Automatic correction of performance drift under acquisition shift in medical image classification"

Comment 2.3: I also think that the paper would be improved if the interplay with other shifts was investigated in some way. I agree with the authors that the proposed method probably could be used as an aid in detecting other shifts, and an experiment showing how it behaves in such a scenario would be enlightening. (If there are space issues, I'd suggest moving fig 4 and/or 5 to supplementary, nice experiments but unsurprising results given fig 2 & 3.)

Response: We thank the reviewer for the suggestion. We have now added an experiment to illustrate the behaviour of UPA under prevalence shift. This experiment aims to shed some light on the ability of UPA to detect shifts, even in scenarios where UPA should not be used for correction (as highlighted in our Discussion section under limitations). In practice, any detected shift would need to be followed-up with an investigation to identify the causes of shift, before applying the correction. We would recommend combining UPA as a correction method with methods that have been specifically developed for performance drift detection (for example, see “Failing Loudly: An Empirical Study of Methods for Detecting Dataset Shift” using two-sample statistical tests). The experiment (Supp. Note. 4) together with further clarifications in the Introduction and Discussion sections have been added to the revised manuscript.

Supp. Fig. 5. Additional experiment illustrating UPA’s ability to detect prevalence shift. We simulate the case where there is prevalence shift instead of acquisition shift between the reference and the “deployment” data, using the example of the histopathology task. We sample three disjoint sets from the in-distribution: the reference set (5000 samples), the alignment set (N=5000) and the evaluation set (N=15,000). Here there is no acquisition shift between reference and deployment data (alignment/evaluation set). Instead, we sample alignment and evaluation sets such that they do not exhibit the same prevalence as the reference set (i.e. prevalence shift) and apply UPA to align the predictions. We then plot the Mean Absolute Difference between original and aligned predictions on the evaluation set in function of the prevalence on the shifted data, while keeping the prevalence in the reference set fixed. We plot results over 500 bootstrap samples. There is a clear correlation between the amount of prevalence shift and the observed differences between original and aligned predictions.

Comment 2.4: A detail is that I'd be interested in also seeing the MCC measure as a complement to sensitivity and specificity in the experiments. As sensitivity and specificity have limitations in imbalanced settings, MCC is a good sanity check that there are no unexpected effects.

Response: Thanks for the suggestion. We considered MCC, however, it is not suitable for extremely unbalanced datasets as it relies on absolute numbers of true negatives / positives. This has been studied in detailed in “On the performance of Matthews correlation coefficient (MCC) for imbalanced dataset” (Zhu, Pattern Recognition Letters, 2020), where the authors show that the behaviour of MCC is particularly affected by the imbalance ratio: *“The higher the imbalance ratio, the MCC measurements tend to be more skewed and behave nonlinearly with respect to the linear increases of TP and TN values”*. As such, we think this metric is not too meaningful in the breast screening setting with a prevalence of 1%. For the balanced histopathology task, the observed MCC values over 500 bootstrap samples were: .870 for site S4 (resp. .865 for S5) on the original data, and .870 (resp. 859 for S5), the differences before and after alignment were not significantly different from zero over the bootstrap samples.

But based on the suggestion, and in order to have a compound measure taking into account both sensitivity and specificity, we now report Youden’s Index, before and after UPA, for Scenario 1 (breast and histopathology) below. As expected the index is preserved or improved by UPA. We have now added these results in the manuscript (Table 3, 4) and thank the reviewer for their suggestion.

Table 3. Youden’s Index for scenario 1, deployment to a new site, breast screening.

Dataset	Before correction	After correction
Scanner B	0.295 (0.011)	0.651 (0.021)
Scanner C	0.599 (0.023)	0.594 (0.020)
Scanner D	0.644 (0.024)	0.640 (0.021)

Reported as average over 500 bootstrap samples, with standard deviation in brackets.

Table 4. Youden’s Index for scenario 1, deployment to a new site, histopathology.

Dataset	Before correction	After correction
Site S4	0.868 (0.003)	0.864 (0.005)
Site S5	0.859 (0.006)	0.863 (0.004)

Reported as average over 500 bootstrap samples, with standard deviation in brackets.

Comment 2.5: A minor detail is that in line 307, sens=spec is referred to as an optimal operating point which is not really true, the optimal choice depends on the use case as the authors correctly say in the introduction.

Response: We thank the reviewer for pointing this out. We agree, and have revised this sentence as follows: “We select the balanced operating point, where SEN equals SPC.”

Comment 2.6: To summarize, the paper is well written and has nice contributions. It could be argued that the amount of scientific contribution is a bit limited for Nature Communications, but for me some deepened investigation such as what’s proposed above would take it over the bar.

Response: We are thankful for the valuable suggestions and hope the reviewer agrees that our modifications have improved our manuscript.

Reviewer 3

Comment 3.1: The authors proposed an automatic recalibration method, unsupervised prediction alignment (UPA), that requires no ground truth annotations and only limited amounts of unlabelled example images from the shifted data distribution. The method is intended to correct for performance drift in AI software caused by changes in data acquisition such as the replacement of scanner hardware or updates to the image processing software. They conducted simulation experiments to demonstrate that UPA can restore the expected performance under several scenarios of image acquisition shift.

The UPA method was relatively simple and could be automated. The authors expected it to be applicable if the data distribution shift was caused by acquisition shift while the population and prevalence remained stable. A major issue is that since the method was applicable only to data acquisition shift and might cause unwanted calibration errors if the data distribution shift was caused by other types of shifts, it is unclear how the method could avoid being mistakenly applied to other types of shifts if it was implemented as an automated procedure in the clinical workflow.

Response: Thanks for raising this point. We have revised our Discussion section and added the following to address this concern:

In practice, we would envision UPA to be used alongside comprehensive monitoring of the input data including meta information about the patient population. This would enable the detection and flagging of unexpected changes in the patient population. Monitoring of base demographics is already standard practice in screening programmes. This could be complemented with automated monitoring tools, for example using auxiliary AI models, to predict patient and data characteristics from the input images (e.g., age, breast density, image quality, etc.) and methods that have been specifically developed for drift detection. (Rabanser et al. 2019). The detection of shifts is an important aspect of continuous performance monitoring. Note that UPA does not have to be applied in real-time, meaning for deployment it may be reasonable to implement a time delay (say a few days) between the detection and the correction of the distribution shift. This would facilitate a human-in-the-loop inspection of whether other sources may have contributed to the detected shift.

Comment 3.2: Another major issue is that the authors assumed that the data distribution shift due to, e.g., a scanner change, could be corrected by a simple “piecewise linear cumulative distribution matching”, which may not be valid in general. For example, mammography systems from different vendors used different target/filter, imaging techniques, detector, and post-processing techniques. An AI software trained with images from one type of scanners cannot be simply applied to another type, or the ROC curve and AUC will change even if the patient population and disease prevalent remain the same. Furthermore, even with the same scanner, acquisition drift could degrade the image quality and cause a degradation in the ROC curve and AUC of the AI software. No simple matching of the AI output distribution can bring

the sensitivity and specificity to the same levels as the reference. As such, the applicability of the proposed UPA will be limited to very few special situations. The use a relatively broad term to describe a very narrow situation is misleading although it was mentioned at the end (Discussion: “acquisition shift primarily induces a shift in the model predictions invalidating the selected operating point...”).

Response: We would like to highlight that the scenarios presented in this paper are in fact of the nature the reviewer mentions where models were trained on one type of imaging system (in case of mammography) or imaging technique (in case of histopathology) and the performance of those models did indeed generalise “out-of-the-box” in terms of ROC-AUC to different types of systems (including different mammography vendors) and techniques. It is therefore not correct that models trained on one system cannot be applied to another type of system. While the reviewer is correct that models are not guaranteed to generalise to new settings, in practice this is often the case (as we show with our different real-world examples). Deep learning systems are strong feature extractors, meaning that once features are learned to discriminate data into relevant classes (e.g., malignant versus non-malignant mammographic images), this ability is often preserved across different datasets. However, the discrimination threshold may be severely shifted, and thus, re-calibration is required to recover the expected performance in terms of sensitivity/specificity trade-off. We believe that our use cases on two very different imaging applications demonstrate that UPA is not limited to a few special situations, but applicable more widely to the general problem of acquisition shift whenever our two main assumptions are satisfied i.e. (i) ROC-AUC is preserved across domains, (ii) prevalence is preserved.

Regarding the applicability of UPA, please see the revised Discussion section which now discusses in more detail the recommended practical use of UPA in deployment.

specific comments:

Comment 3.3: Introduction, the authors correctly pointed out that “The topic of performance monitoring is a pressing matter for the practical deployment of machine learning models”. However, it is unclear how the proposed method could automatically detect performance drift. The simulated scenarios in their study assumed that a new/different scanner would be introduced or an old one would be replaced, or a software upgrade would be performed, etc. With such known events, the UPA could be invoked to make corrections, if appropriate. In real-world clinical practice, it is difficult to detect any shift in the AI output data distribution unless a QA monitoring procedure is implemented to collect AI performance data with or without respect to some “reference truth” statistically over time.

Response: Please see our response to Comment 3.1 and our revised Discussion regarding the practical use of UPA in deployment. Indeed, we would envision UPA to be used alongside other monitoring including comprehensive quality control (which is often already in place in clinical applications such as breast cancer screening). In addition, we would like to highlight our newly added experiments (see Supp. Note 4, Supp. Fig. 5) on UPA’s ability to detect other types of distribution shift such as prevalence shift. Thus, UPA, possibly in combination with other drift detection methods (see Rabanser et al. 2019), is capable of detecting drifts over time. We agree that the detection and correction steps may need to be decoupled in practical use cases, as discussed in our revised Discussion section.

Comment 3.4: As commented above, it is unclear how the proposed method could automatically detect model performance shift and determine whether the shift was within those the UPA could handle. In general, many factors can cause shifts in the image quality and image characteristics and the different types of shifts can mix together. The statement “there is an unmet need for methods allowing to automatically detect and correct model performance drift in the absence of any diagnostic information ... This is precisely the focus of this work.” appears to imply that the proposed method and study could meet this need. Please clarify.

Response: We hope our revised Discussion section, where we discuss the practical use of UPA, its ability to detect shifts including prevalence shift, and the possible combination of other monitoring tools and shift detection algorithms, addresses the reviewer’s concern. We believe our statement remains valid and appropriate in the context of our work.

Comment 3.5: Datasets, “all other evaluation sets are screening datasets from three sites in the UK and four sites in Hungary representative of the real-world populations in the respective national breast cancer screening programmes.”: Please specify how the seven sites were grouped into the four datasets described as scanners A-D in Table 1, and whether each dataset actually were collected with a single scanner type, and if so, what type for each set. As described, the seven sites came from two national screening programs from two countries, how did the authors determine that data distribution shift were caused by scanner difference or software upgrade only? One would expect that population shift, for example, could be a major factor between datasets from the two countries. It is important to be specific about the characteristics of these datasets because they were used as examples to demonstrate the proposed method could work well. The results of the experiments gave the impression that the method was quite versatile and applicable to international multi-site data from four different types of scanners.

Response: Thanks for the comment, we agree that this is important information as it highlights the strength of our approach. We have now added the requested information in Table 1 with details about the scanner manufacturers, and how the four datasets relate to the two screening programmes. We can confirm that each dataset corresponds to images from specific types of scanners from four different manufacturers. The dataset described as ‘scanner A’ corresponds to Hologic (from the OPTIMAM database), ‘scanner B’ corresponds to IMS Giotto (which is the manufacturer used in all four Hungarian sites which are grouped together), ‘scanner C’ to Siemens, and ‘scanner D’ to GE. The images from scanners A, C, and D were collected at UK sites, and the datasets were grouped by scanner manufacturer. Thus, we do believe that our results demonstrate the versatile nature of UPA in a real-world setting including international, multi-site data from four different types of scanners. While we do report prevalence across the datasets and two countries, which is largely similar and in line with expectations for breast cancer screening programmes, the reviewer is right that other population shifts may be present, in particular, between the UK and Hungarian population. However, what is important here is that UPA can recover the desired sensitivity/specificity trade-off in all tested cases despite these (unknown) differences.

Comment 3.6: Discussion, last paragraph “Additionally, ...”: The discussion here is general and important for deployment of medical AI to clinical use; however, it is not specific to the

study or method proposed in this paper. The general justifications have been discussed in the Introduction. Concluding the study with this discussion is too broad and out of scope.

Response: Thanks for the feedback. We hope that our more detailed discussion regarding the practical use of UPA, the underlying assumptions, and the limitations addresses this concern. We do, however, feel there is value in concluding on a more general outlook on medical AI, the importance of monitoring and auditing, and the implications of methods such as UPA in the context of medical device regulation. The FDA and other regulatory bodies are actively investigating and seeking input on dealing with AI-based devices which continuously learn and/or update over time. Our work may contribute to this discussion by proposing a practical approach for AI monitoring and performance correction. We decided to keep this paragraph, but are happy to revise if the reviewers and editor feel it is out of scope.

REVIEWERS' COMMENTS

Reviewer #1 (Remarks to the Author):

The new version of the manuscript has adequately addressed my points. In particular the new results on different Sens / Spec tradeoffs and the results of prevalence shift. I also appreciate the updated discussion on generalisability across diverse models.

I believe that this manuscript is now ready for publication and offers a valuable addition to the area of shift detection / updating in healthcare.

Reviewer #2 (Remarks to the Author):

The authors have thoroughly considered the review comments and made good changes and additions that have further increased the quality of the manuscript. I recommend that it is accepted for publication.

Reviewer #3 (Remarks to the Author):

The revised manuscript has clarified most of the issues, especially the limitations of the application of the UPA. However, the appropriateness of the last paragraph of the Discussion section is still questionable.

Specific comments:

1. The last paragraph of the Discussion section: the addition of the new sentence may tie the first part of the paragraph a little to the study. However, the discussion "... especially when considering an unsupervised modification to parameters" indicates that they suggested the application of UPA to automatic correction of these AI systems. In addition, in the latter part "Modern AI software as a medical device regulations include the option of "change protocols" to allow AI devices to improve over

time but must be negotiated and agreed upon with the regulators and include appropriate human oversight in deployment.” is more explicit about the possible change in the performance of these AI systems. If an AI device is allowed to “improve over time” or undergo “unsupervised modification to parameters”, its ROC-AUC would increase (or change) over time. As the authors stated, one of the main assumptions that the UPA method depends on is that the ROC-AUC stays the same across domains or data sets, which is contradictory to the expected increase (or change) in performance for the AI systems discussed in this paragraph.

2. The first part of the last sentence “Thorough validation of UPA systems influencing the performance of the device, comprehensive post-market systems, ...” is not understandable.

Response to reviewers

Reviewers 1 & 2

Comment 1.1: The new version of the manuscript has adequately addressed my points. In particular the new results on different Sens / Spec tradeoffs and the results of prevalence shift. I also appreciate the updated discussion on generalisability across diverse models. I believe that this manuscript is now ready for publication and offers a valuable addition to the area of shift detection / updating in healthcare.

Comment 2.1: The authors have thoroughly considered the review comments and made good changes and additions that have further increased the quality of the manuscript. I recommend that it is accepted for publication.

Response: Many thanks for taking the time to assess our revised manuscript. We are very pleased that the reviewers are satisfied with our changes. We appreciate the valuable and constructive feedback provided earlier.

Reviewer 3

The revised manuscript has clarified most of the issues, especially the limitations of the application of the UPA. However, the appropriateness of the last paragraph of the

Discussion section is still questionable.

Specific comments:

Comment 3.1: The last paragraph of the Discussion section: the addition of the new sentence may tie the first part of the paragraph a little to the study. However, the discussion "... especially when considering an unsupervised modification to parameters" indicates that they suggested the application of UPA to automatic correction of these AI systems. In addition, in the latter part "Modern AI software as a medical device regulations include the option of "change protocols" to allow AI devices to improve over time but must be negotiated and agreed upon with the regulators and include appropriate human oversight in deployment." is more explicit about the possible change in the performance of these AI systems. If an AI device is allowed to "improve over time" or undergo "unsupervised modification to parameters", its ROC-AUC would increase (or change) over time. As the authors stated, one of the main assumptions that the UPA method depends on is that the ROC-AUC stays the same across domains or data sets, which is contradictory to the expected increase (or change) in performance for the AI systems discussed in this paragraph.

Response: We thank the reviewer for appreciating the changes made during our previous revision to our manuscript. We would like to clarify that in this manuscript we measure "performance" by clinically relevant metrics: sensitivity and specificity. As we demonstrate in our first figure, the clinical performance of an algorithm does not solely depend on its measured ROC-AUC but is also highly dependent on choosing the appropriate classification

threshold. As such an algorithm can “improve” in terms of expected sensitivity / specificity while remaining at the same ROC-AUC.

We have added a clarification in the Discussion to the sentence highlighted by the reviewer which now reads “[...] to allow AI devices to improve over time with respect to clinically relevant metrics [...]”.

Comment 3.2: The first part of the last sentence “Thorough validation of UPA systems influencing the performance of the device, comprehensive post-market systems, ...” is not understandable.

Response: Many thanks for pointing this out. We have now revised the sentence to read “Thorough validation of processes influencing the performance of a medical device, as well as comprehensive post-market surveillance and continuous model monitoring are critical to the success of this approach in a regulated environment.”